# 2-Alkyl-4-hydroxyquinolines from a Marine-Derived *Streptomyces* sp. Inhibit Hyphal Growth Induction in *Candida albicans*

**DOI:** 10.3390/md17020133

**Published:** 2019-02-22

**Authors:** Heegyu Kim, Ji-Yeon Hwang, Beomkoo Chung, Eunji Cho, Suhyun Bae, Jongheon Shin, Ki-Bong Oh

**Affiliations:** 1Department of Agricultural Biotechnology, College of Agriculture and Life Sciences, Seoul National University, Seoul 08826, Korea; hqhqeori@snu.ac.kr (H.K.); beomkoo01@snu.ac.kr (B.C.); eunji525@snu.ac.kr (E.C.); hyunature@snu.ac.kr (S.B.); 2Natural Products Research Institute, College of Pharmacy, Seoul National University, Seoul 08826, Korea; yahyah7@snu.ac.kr

**Keywords:** 2-alkyl-4-hydroxyquinolines, marine actinomycete, *Streptomyces* sp., *Candida albicans*, morphogenesis, cAMP-Efg1 pathway

## Abstract

Four 2-alkyl-4-hydroxyquinoline derivatives (**1**–**4**) were isolated from a semisolid rice culture of the marine-derived actinomycete *Streptomyces* sp. MBTG13. The structures of these compounds were elucidated by a combination of spectroscopic methods, and their data were in good agreement with previous reports. Compounds **1** and **2** exhibited weak to moderate antibacterial activity against pathogenic bacteria. Unexpectedly, we found that compound **1** acted as a potent inhibitor of hyphal growth induction in the dimorphic fungus *Candida albicans*, with an IC_50_ value of 11.4 μg/mL. Growth experiments showed that this compound did not inhibit yeast cell growth, but inhibited hyphal growth induction. Semi-quantitative reverse transcription (RT)-PCR analysis of hyphal-inducing signaling pathway components indicated that compound **1** inhibited the expression of mRNAs related to the cAMP-Efg1 pathway. The expression of *HWP1* and *ALS3* mRNAs (hypha-specific genes positively regulated by Efg1, an important regulator of cell wall dynamics) was significantly inhibited by the addition of compound **1**. These results indicate that compound **1** acts on the Efg1-mediated cAMP pathway and regulates hyphal growth in *Candida albicans*.

## 1. Introduction

*Candida albicans* is an important human fungal pathogen that can reversibly transition between two distinct morphological forms: yeast and filamentous hypha [1,2]. Furthermore, the morphological transition ability of the organism contributes to its virulence [3], and hyphal development is closely associated with the dissemination of, and tissue invasion by, *C. albicans*. Hyphal development in *C. albicans* is triggered by various in vitro environmental signals such as neutral pH, nutrient-poor media, high temperatures, a high ratio of CO_2_, and serum exposure [4]. In addition to environmental signals, the morphological transition of *C. albicans* is controlled by a complex network of signaling pathways, including the Cph1-mediated MAPK pathway and the Efg1-mediated cAMP pathway. Ras1 likely acts upstream of both pathways as an important regulator of hyphal development [2].

Quinoline alkaloids possess a broad range of biological activities such as anticancer, antimicrobial, antimalarial, and anti-inflammatory activities, and they are found in various organisms, including higher plants [5,6,7], fungi [8,9], and bacteria [10,11,12] such as marine-derived actinomycetes [13]. Among these compounds, 2-alkyl-4-hydroxyquinolines (4-hydroxy-2-alkylquinolines) are frequently found in various strains of *Pseudomonas* spp. [11,14,15,16,17], and they are known as quorum-sensing molecules, involved in cell-to-cell communication [18].

In our continuing search for bioactive secondary metabolites from marine-derived actinomycetes, we characterized a strain, MBTG13, collected from marine sediment from Jeju Island, Republic of Korea, identified as *Streptomyces* sp. by its 16S rDNA. An organic extract of a semisolid rice culture of this strain exhibited weak antibacterial activity (minimum inhibitory concentration 64 μg/mL) against two pathogenic bacteria (*Staphylococcus aureus* and *Salmonella enterica*). Activity-guided separation of the extract employing diverse chromatographic methods led to the isolation of 2-alkyl-4-hydroxyquinoline derivatives **1**–**4**. In this study, we discovered that these compounds, especially compound **1**, were inhibitors of filamentous growth induction in the dimorphic fungus *C. albicans*. This is the first study to identify the mechanistic effect of this compound on *C. albicans* morphogenesis.

## 2. Results

### 2.1. Taxonomy and Phylogenetic Analysis of MBTG13

The 16S rDNA of strain MBTG13 was amplified by polymerase chain reaction (PCR) and sequenced. After a basic logic alignment search tool (BLAST) sequence comparison, strain MBTG13 showed 99% identity to *Streptomyces luridiscabiei* (GenBank accession number: NR_025155). Thus, this strain was designated as *Streptomyces* sp. MBTG13 (GenBank accession number: MK408429). The phylogenetic tree that was generated by the neighbour-joining and maximum likelihood methods based on the 16S rDNA sequence revealed the evolutionary relationships of strain MBTG13 with a group of known Streptomyces species (Figure 1).

### 2.2. Isolation and Structural Elucidation of Compounds ***1**–**4***

Strain MBTG13 was cultured in semisolid rice medium and extracted with MeOH and CH_2_Cl_2_. After evaporation of the solvents, the combined extract was separated by solvent-partitioning followed by reversed-phase C_18_ vacuum flash chromatography and preparative high-pressure liquid chromatography (HPLC) to yield four compounds. By combined spectroscopic analyses, including ^1^H, ^13^C NMR, two-dimensional (2D) NMR spectral analyses (COSY, HMQC, and HMBC), and UV data, compounds **1**–**4** were identified as 2-*n*-heptyl-4-hydroxyquinoline (**1**) [11,14], 2-*n*-heptyl-4-hydroxyquinoline-*N*-oxide (**2**) [11], 2-*n*-octyl-4-hydroxyquinoline (**3**) [17], and 3-*n*-heptyl-3-hydroxy-1,2,3,4-tetrahydroquinoline-2,4-dione (**4**) [11] (Figure 2). The spectroscopic data for these compounds were in good agreement with previous reports.

### 2.3. Antimicrobial Activity of Compounds ***1**–**4***

Because compound **1** was initially reported as an antibacterial compound [14], the antimicrobial activities of the isolated compounds were first evaluated against phylogenetically diverse pathogenic bacterial strains, including *Staphylococcus aureus* ATCC25923, *Enterococcus faecalis* ATCC19433, *Enterococcus faecium* ATCC19434, *Salmonella enterica* ATCC14028, *Klebsiella pneumoniae* ATCC10031, and *Escherichia coli* ATCC25922, using ampicillin and tetracycline as positive control compounds (Table 1). Compound **1** displayed weak antibacterial activity against *S. aureus* ATCC 25923, *E. faecalis* ATCC19433, and *E. coli* ATCC25922, with minimum inhibitory concentration (MIC) values of 128 μg/mL, 128 μg/mL, and 64 μg/mL, respectively. Compound **2** broadly inhibited most of the tested bacterial pathogens, except *K. pneumoniae* and *E. coli*, with MIC values in the range of 16–32 μg/mL. The antifungal activities of compounds **1**–**4** were also evaluated against pathogenic fungal strains, including *Candida albicans* SC5314, *Aspergillus fumigatus* HIC6094, *Trichophyton rubrum* NBRC9185, and *Trichophyton mentagrophytes* IFM40996, using amphotericin B as a positive control compound. However, compounds **1**–**4** did not exhibit inhibitory activity against the tested fungi (MIC > 128 μg/mL).

### 2.4. Effects of Compounds ***1**–**4*** on C. albicans Morphogenesis

The effects of isolated compounds **1**–**4** on *C. albicans* SC5314 growth and morphogenesis were evaluated. First, to evaluate the effects of these compounds on *C. albicans* yeast growth, the cells were grown in glucose salt (GS) medium supplemented with 100 µg/mL of test compound at 28 °C, and the optical density at 660 nm (OD_660_) of each sample was measured at each specific time interval. Compounds **1**–**4** at 100 µg/mL did not inhibit yeast cell growth in *C. albicans* (Figure 3a). To evaluate the effects of compounds **1**–**4** on the hyphal growth of *C. albicans*, approximately 5 × 10^6^ cells/mL were added to GS medium supplemented with the test compounds at different concentrations and incubated at 37 °C. At each time point, the morphology of approximately 200 cells was determined by light microscopy (Figure 3b). Under these conditions, >90% of *C. albicans* cells converted to the hyphal form after 4 h of incubation. Cultures treated with compounds **1**–**4** exhibited concentration-dependent inhibition of the hyphal form of *C. albicans*. Among these compounds, compound **1** exhibited potent inhibitory activity, with an IC_50_ of 11.4 µg/mL. Oxidation of 4-hydroxyquinoline moiety (**2**) (Figure 2) led to a loss of inhibitory activity compared to compound **1**. With regard to the effect of alkyl chain length, for a given compound **1**, extension of the chain by one methyl unit (**3**) resulted in a dramatic decrease in inhibitory effect upon hyphal formation at the highest concentration tested (50% at 100 µg/mL). From these results, it is apparent that the inhibitory potency and selectivity of 2-alkyl-4-hydroxyquinoline derivatives (**1**–**4**) are sensitively dependent upon the chain length, as well as to substitutions on the 4-hydroxyquinoline template. Taken together, these data suggest that 2-alkyl-4-hydroxyquinoline derivatives regulate the hyphal formation process of *C. albicans* without interfering with its yeast form proliferation.

### 2.5. Analysis of Gene Expression Related to Hypha-Inducing Signaling Pathways

In *C. albicans*, hyphal development is mainly regulated by the mitogen-activated protein kinase (MAPK) and cyclic AMP-protein kinase A (cAMP-PKA) pathways, and active Ras1 is required for the regulation of both pathways [2]. The transcription factors Efg1 and Cph1 are activated by distinct upstream signaling pathways. In the case of Efg1, the pathway is based on cAMP, while in the case of Cph1 the pathway depends on a MAPK signaling pathway, with Ras1 stimulating both pathways. To investigate the effect of compound **1** on hypha-inducing signaling pathways in *C. albicans*, the mRNA expression levels of *CPH1*, *EFG1*, *GAP1*, and *HWP1* were determined using gene-specific primers (Table 2).

Semi-quantitative reverse transcription (RT)-PCR showed that the mRNA expression of *CPH1* and *EFG1* was not repressed in compound **1**-treated cells (Figure 4a). The transcript level of *GAP1*, which encodes a general amino acid permease and is positively regulated by the transcription factor Cph1 [4] was also unaffected. Importantly, a complete loss of *HWP1* mRNA expression occurred with 100 µg/mL of compound **1**. *HWP1*, which encodes a glycosylphosphatidylinositol-anchored cell wall protein [19], is a downstream component of the cAMP-dependent PKA pathway and is positively regulated by the transcription factor Efg1, which is an important regulator of cell wall dynamics [20]. We next investigated the relationship between *HWP1* transcript levels and hypha formation in *C. albicans* cultures grown in GS medium treated with increasing concentrations of compound **1** at 37 °C for 2 h. Figure 4b shows that the level of *HWP1* transcript was undetectable in yeast cells grown in GS medium at 28 °C for 2 h. In contrast, under hyphal growth-inducing conditions, the *HWP1* transcript level was increased with a decrease in concentration of compound **1**, mainly due to the predominant growth of the hyphal population. These results suggest that compound **1** functions on the cAMP-Efg1 pathway, but not the MAPK-Cph1 pathway.

Gene-specific studies and genome-wide analyses have revealed only a small number of hypha-specific genes in *C. albicans* [21]. These include *ALS3*, *ECE1*, *HGC1*, *HWP1*, *HYR1*, *RBT1*, and *RBT4*. The most highly expressed gene encodes *HWP1,* which is one of a small set of hypha-specific genes in *C. albicans* that include *ALS3* (encoding agglutinin-like protein 3) and *ECE1* (encoding cell elongation protein 1). Among them, *HWP1* and *ALS3* encode adhesins and are activated by the transcription regulator Efg1 during hypha formation [22]. Thus, we investigated whether compound **1** also affected *HWP1* and *ALS3* mRNA levels (Figure 4c). In the absence of compound **1**, *HWP1* and *ALS3* transcripts were undetectable at time zero, but they were strongly expressed from 20 to 80 min. Importantly, compound **1** (100 µg/mL) dramatically reduced the magnitude of *HWP1* and *ALS3* mRNA expression in compound **1**-treated cells, when compared with untreated cells.

## 3. Discussion

A chemical investigation of a semisolid rice culture extract of the marine-derived actinomycete *Streptomyces* sp. MBLG13 led to the isolation of 2-alkyl-4-hydroxyquinoline derivatives (**1**–**4**). Their structures were elucidated based on spectroscopic data and were in good agreement with previous reports. Compounds **1** and **2** exhibited weak to moderate antibacterial activity against pathogenic bacteria. Compound **1** was earlier isolated from *Pseudomonas aeruginosa* and had antibacterial activity against Gram-positive bacteria [14]. In addition, compounds **1**, **2**, and **4** were isolated from *Pseudomonas methanica* KY4634 as 5-lipoxygenase inhibitors [11]. Recently, it was reported that compounds **1**–**3** exhibited antimalarial activity against *Plasmodium falciparum* [17]. In this study, we found that compounds **1** and **3** inhibited an essential *C. albicans* virulence factor, especially against hyphal growth, without inhibiting yeast cell growth. Our initial assumption was that these compounds were inhibitors of hyphal growth induction in *C. albicans* due to their structural analogies to known quorum sensing signal molecules, including farnesoic acid and farnesol [23,24]. These molecules are secreted into the medium as the cells proliferate and are involved in morphogenesis. Thus, we examined whether these compounds acted as inhibitors of the morphological transition in *C. albicans* under hypha-inducing conditions. Our results reveal that 2-alkyl-4-hydroxyquinoline derivatives regulate the hyphal formation process of *C. albicans* without interfering with its yeast form proliferation. The inhibitory potency and selectivity of these compounds are sensitively dependent upon the chain length and to substitutions on the 4-hydroxyquinoline template.

In *C. albicans*, the positive regulation of hypha-specific gene expression is mainly mediated through the MAPK and cAMP-PKA signaling pathways [1,2]. Transcription factors Efg1 and Cph1 are activated by distinct upstream signaling pathways. In the case of Efg1, the pathway is based on cAMP, while in the case of Cph1 the pathway depends on MAPK signaling. To investigate the networks by which the morphogenesis of *C. albicans* is controlled by compound **1**, we compared the expression levels of these major hypha-inducing signaling pathway components in compound **1**-treated cells with untreated control cells. When cells were exposed to compound **1**, the mRNA expression of *HWP1* and *ALS3*, hypha-specific genes that are positively regulated by an important regulator of cell wall dynamics (Efg1), were significantly inhibited by the addition of compound **1**. In contrast, the transcription of *CPH1*, *EFG1*, and *GAP1* mRNA was unchanged.

Efg1 and Cph1 are transcription factors whose activity is regulated on a post-transcriptional level. Threonine-206, a phosphorylation site for protein kinase A (PKA) within an Efg1p domain, is essential to promote hyphal induction by environmental factors [25]. Previous studies have examined the role of Efg1 and Cph1 in the repression of hyphal growth by farnesol, a related compound. It has been found that even though these proteins are clearly involved in mediating the farnesol effect, there is no significant change in *EFG1* or *CPH1* mRNA [26,27]. In our results, *EFG1* mRNA levels are regulated during hyphal development, but they were not affected by compound **1**, since the magnitude of the changes were similar in the presence and absence of compound **1** (Figure 4a). These results are consistent with those of Kebaara et al. [26]. Together with our results, this suggests that compound **1** does not regulate *EFG1* mRNA levels, but at this time, we cannot exclude the possibility that post-translational regulation of Efg1 is affected by compound **1**. This study is the first report demonstrating inhibitory activity of *C. albicans* filamentation by compounds **1**–**4**, and suggests that the 2-alkyl-4-hydroxyquinoline class of compounds may be valuable as antifungal agents to suppress virulence in *C. albicans*.

## 4. Materials and Methods

### 4.1. General Experimental Procedures

^1^H, ^13^C, and 2D NMR spectra were measured using a Bruker Avance 600 MHz spectrometer at the National Center for Interuniversity Research Facilities (NCIRF) located in Seoul National University. Mass spectrometric data were obtained at the Korea Basic Science Institute (Daegu, Republic of Korea) and were acquired using a JMS 700 mass spectrometer (JEOL, Tokyo, Japan) using meta-nitrobenzyl alcohol as a matrix for fast atom bombardment mass spectrometry. HPLC was performed using a Spectrasystem p2000 (Thermo Scientific, Waltham, MA, USA) equipped with a Spectrasystem RI-150 refractive index detector. All organic solvents used in the experiment purchased from Fisher Scientific (FairLawn, NJ, USA).

### 4.2. Taxonomic Identification of the Producing Microorganism

The bacterial strain, *Streptomyces* sp. (strain number MBTG13), was isolated from a marine sediment sample from the shoreline of Jeju Island, Republic of Korea. The strain was identified using standard molecular biological protocols by DNA amplification and sequencing of the internal transcribed spacer region. The genomic DNA extraction was performed using an i-Genomic BYF DNA Extraction Mini Kit (Intron Biotechnology, Seoul, Republic of Korea) according to the manufacturer’s protocol. The nucleotide sequence of MBTG13 has been assigned as accession number MK408429 in the GenBank database. The 16S rDNA sequence of this strain showed 99% identity with *S. luridiscabiei* (GenBank accession number: NR_025155) based on comparison in basic local alignment search tool (BLAST).

### 4.3. Fermentation and Isolation

Strain MBTG13 was cultivated on a yeast extract–peptone–glycerol (YPG) agar plate (5 g of yeast extract, 5 g of peptone, 10 g of glucose, and 16.0 g of agar in 1 L of artificial seawater) for 5 days at 28 °C. It formed brown-yellow colonies with white spore on plates. Agar plugs (1 cm × 1 cm, five pieces each) were inoculated into 100 mL of YPG media for 5 days to obtain seed broth. For the purpose of large-scale fermentation, seed cultures separately were transferred to 24 flasks, each containing 2 g of peptone, 2 g of yeast extract, and 200 g of rice with 200 mL of artificial seawater.

The fermentation in rice media was conducted under static conditions for 6 weeks at 28 °C. A total of 510 g of rice fermentation culture was repeatedly extracted by MeOH (1 L × 3) and CH_2_Cl_2_ (1 L × 3). The organic solvents were evaporated to dryness under reduced pressure to gain 8.5 g of total extracts. A total extract (8.5 g) was successively partitioned between *n*-BuOH (1.6 g) and H_2_O (6.2 g). The *n*-BuOH fraction was repartitioned using H_2_O–MeOH (15:85) (1.1 g) and *n*-hexane. The H_2_O–MeOH fraction was subjected to C_18_ reverse-phase vacuum flash chromatography using a gradient of MeOH and H_2_O to give seven fractions (five fractions in the gradient, H_2_O–MeOH, from 0:100 to 100:0, acetone, and finally EtOAc). On the basis of the results of biological analyses, the fraction eluting with H_2_O–MeOH (20:80) (1.1 g) was separated and purified by semi-preparative reverse-phase HPLC (Agilient C_18_ column, 10.0 × 250 mm; H_2_O–MeOH, 30:70 with 0.1% trifluoroacetic acid; 2.0 mL/min) to obtain compounds **1** (*t*R: 16 min, 24.2 mg), **2** (*t*R: 24 min, 3.5 mg), **3** (*t*R: 26 min, 2.5 mg), and **4** (*t*R: 31 min, 2.7 mg).

### 4.4. Antibacterial Activity Assay

The antibacterial activity assay was carried out according to the Clinical and Laboratory Standards Institute (CLSI) method [28]. Gram-positive bacteria (*S. aureus* ATCC25923, *E. faecalis* ATCC19433, *E. faecium* ATCC19434) and Gram-negative bacteria (*K. pneumoniae* ATCC10031, *S. enterica* ATCC14028, *E. coli* ATCC25922) were cultured overnight in Mueller Hinton broth (MHB) at 37 °C. The cells were collected by centrifugation, and washed two times with sterile distilled water. Each test compound was dissolved in dimethyl sulfoxide (DMSO) and diluted with MBH to prepare serial twofold dilutions in the range of 0.06–128 μg/mL. The final DMSO concentration was maintained at 1% by adding DMSO to the medium as guided by the CLSI [28]. In each well of a 96-well plate, 190 μL of MBH containing the test compound was mixed with 10 μL of the broth containing approximately 10^6^ colony-forming units (cfu)/mL of test bacterium (final concentration: 5 × 10^4^ cfu/mL) adjusted to match the turbidity of a 0.5 MacFarland standard. The plates were incubated for 24 h at 37 °C. The MIC was defined as the lowest concentration of test compound that prevented cell growth. Ampicillin and tetracycline were used as reference compounds.

### 4.5. Antifungal Activity Assay

The antifungal activity assay was performed in accordance with the guidelines in CLSI document M38 [29]. *C. albicans* SC5314 was cultured on potato dextrose agar (PDA). After incubation for 48 h at 28 °C, yeast cells were harvested by centrifugation and washed twice with sterile distilled water. *Aspergillus fumigatus* HIC6094, *T. rubrum* NBRC9185, and *T. mentagrophytes* IFM40996 were plated on PDA and incubated for 2 weeks at 28 °C. Spores were harvested and washed two times with sterile distilled water. Stock solutions of the compound were prepared in DMSO. Each stock solution was diluted in RPMI 1640 broth (Difco, Livonia, MI, USA) with the concentration range of 0.06–128 μg/mL. The final DMSO concentration was maintained at 1% by adding DMSO to the broth. Harvested cells were adjusted to 0.5 MacFarland standard in RPMI 1640 broth and the diluted cells were added to each well of a 96-well plate with test compound solution. The final inoculum concentration was 10^4^ cells/mL in each well. The plates were incubated for 24 h (for *C. albicans*), 48 h (for *A. fumigatus*), and 96 h (*for T. rubrum* and *T. mentagrophytes*) at 37 °C. The MIC value was determined as the lowest concentration of test compound that fully inhibited cell growth. A culture with DMSO (1%) was used as a solvent control, and a culture supplemented with amphotericin B was used as a positive control.

### 4.6. Candida albicans Strain and Growth Medium

*C. albicans* strain SC5314 was used for most experiments. To prepare cells, *C. albicans* cultures were grown at 28 °C with shaking in YPD (1% yeast extract, 2% peptone, and 2% dextrose) medium to the exponential phase, harvested by centrifugation, washed, resuspended in distilled water, and stored for 48 h at 4 °C before use. The hypha-inducing media used in this study was composed of GS medium (5 g of glucose, 0.26 g of Na_2_HPO_4_·12H_2_O, 0.66 g of KH_2_PO_4_, 0.88 g of MgSO_4_·7H_2_O, 0.33 g of NH_4_Cl, and 16 µg of biotin per L) [23].

### 4.7. Candida albicans Growth and Morphological Transition Assay

*C. albicans* SC5314 cells were pregrown in YPD medium at 28 °C up to the exponential phase, washed, and starved in water for 48 h at 4 °C. Stock solutions of compounds **1**–**4** (4 mg/mL) were prepared in DMSO and added to test media at the prescribed concentrations (final DMSO concentration: 1%). For the yeast cell growth assay, 1 × 10^6^ cells/mL were inoculated in GS medium containing test compounds at different concentrations, and the cultures were incubated at 28 °C for 24 h. The number of cells was calculated by measuring the OD_660_ of vigorously vortexed cultures at designated time points (every 4 h until 24 h). For the hyphal formation assay, approximately 5 × 10^6^ cells/mL were added to GS medium containing test compounds at different concentrations and incubated at 37 °C for 4 h. GS medium with 1% DMSO was used as a negative control. At each time point, the morphology of the cells was determined by light microscopy. A minimum of 200 cells were counted for each sample. Assays were carried out three times, each with three replicates.

### 4.8. Gene Expression Analysis

*C. albicans* strain SC5314 was cultured in YPD medium at 28 °C on rotary shakers for 24 h, harvested by centrifugation, washed, resuspended in distilled water, and incubated at 4 °C for 48 h before use. GS medium containing compound **1** (25, 50, and 100 μg/mL) was added to *C. albicans* cells, and each sample was incubated at 37 °C for 2 h. Extraction of total RNA was carried out using an RNeasy Mini Kit (Qiagen, San Diego, CA, USA) and 1 μg of extracted total RNA was reverse transcribed into cDNA using the Superscript III First-Strand Synthesis System (Invitrogen, Carlsbad, CA, USA) with oligo (dT)_20_ (50 mM) primer. Semi-quantitative RT-PCR was conducted with gene-specific primers for major components of the signaling pathways (Table 2). Signaling pathway components are coded as follows: *CPH1* (transcription factor), MAPK pathway; *EFG1* (transcription factor), cAMP-protein kinase A complex (PKA) pathway; *GAP1*, which encodes a general amino acid permease and is positively regulated by Cph1; *HWP1* and *ALS3*, which encode hyphal wall protein 1 and agglutinin-like protein 3, respectively, and are activated by Efg1. The housekeeping gene *GPD1* (encoding glyceraldehyde-3-phosphate dehydrogenase) was used as a loading control [30]. PCR conditions were programmed according to the manufacturer’s instructions: initial denaturation at 98 °C for 5 min, followed by 30 cycles of denaturation at 98 °C for 10 s, annealing at 55 °C for 30 s, and extension at 72 °C for 1 min, with a final extension at 72 °C for 5 min. ImageJ software (National Institutes of Health, Bethesda, MD, USA) was used for densitometric analyses of the mRNA expression levels.

## Figures and Tables

**Figure 1 marinedrugs-17-00133-f001:**
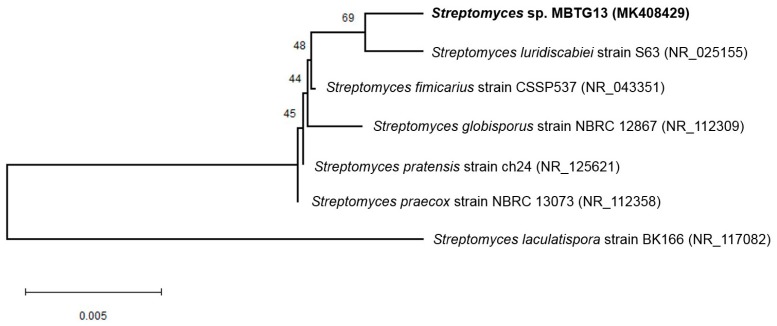
Neighbor-joining phylogenetic tree made by 16S rDNA sequence analysis, showing the position of *Streptomyces* sp. MBTG13 and its closely related phylogenetic neighbors in the MEGA X. Bootstrap was performed with 1000 replicates. The Kimura two-parameter model was used for measuring distance. Bar indicates 0.5% sequence divergence.

**Figure 2 marinedrugs-17-00133-f002:**
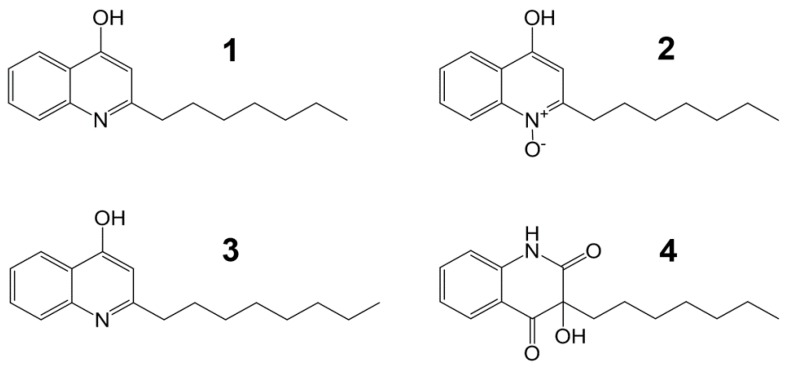
The structures of compounds **1**–**4**.

**Figure 3 marinedrugs-17-00133-f003:**
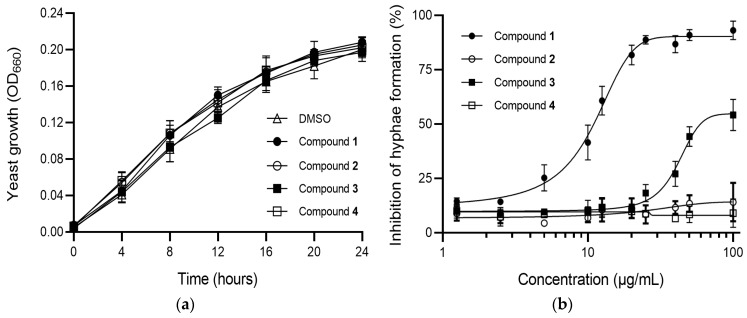
Effects of compounds **1**–**4** on yeast cell growth and hyphal growth induction in *C. albicans* SC5314. (**a**) Effects of compounds **1**–**4** (each 100 μg/mL) on yeast cell growth in *C. albicans*. Glucose salt (GS) medium with 1% dimethyl sulfoxide (DMSO) was used as control. The number of cells at each specific time point at 28 °C was assessed by measuring the optical density absorption at 660 nm (OD_660_). (**b**) Effects of compounds **1**–**4** on hyphal formation in *C. albicans*. Cells (5 × 10^6^ cells/mL) were grown in GS medium containing different concentrations of test compound at 37 °C. At least 200 cells were counted for each sample after 4 h of cultivation. Data are presented as the mean fold changes ± SD of three independent experiments.

**Figure 4 marinedrugs-17-00133-f004:**
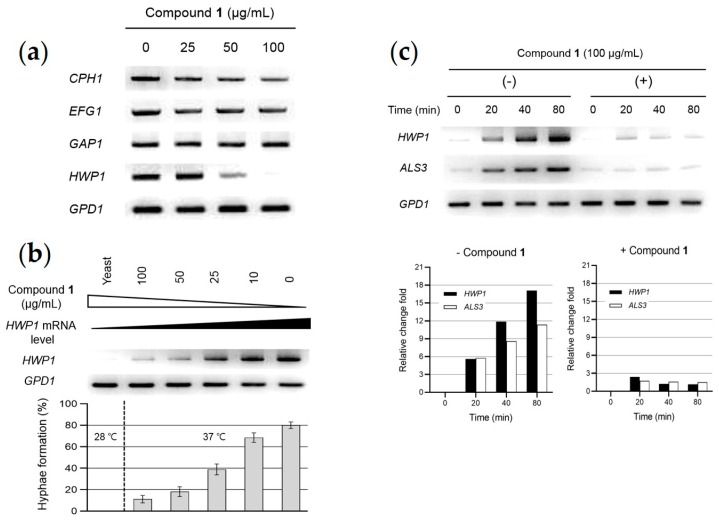
Semi-quantitative reverse transcription (RT)-PCR analysis of mRNAs related to the hyphal-inducing signaling pathway in *C. albicans* SC5314 cells. (**a**) Relative expression of mRNAs related to the hyphal-inducing signaling pathway in *C. albicans*. Cells were incubated in the absence or presence of compound **1** at 37 °C for 2 h, followed by RNA isolation and cDNA synthesis. Semi-quantitative RT-PCR analysis was conducted with gene-specific primers (Table 2). (**b**) Relationship between *HWP1* transcript level and hyphal formation in *C. albicans* cultures grown in GS medium treated with increasing concentrations of compound **1** at 37 °C for 2 h. *C. albicans* yeast cells were grown in GS medium without compound **1** at 28 °C for 2 h. (**c**) Kinetic analysis of hypha-specific *HWP1* and *ALS3* mRNA levels. *C. albicans* cells were incubated in the absence or presence of 100 μg/mL compound **1** at 37 °C in GS medium. Cells were then harvested at 0-, 20-, 40-, and 80-min post-incubation. ImageJ software was used for densitometric analysis of mRNA expression level.

**Table 1 marinedrugs-17-00133-t001:** Results of antimicrobial activity test.

Compound	Minimum Inhibitory Concentration (MIC) (μg/mL)									
	Gram(+) Bacteria			Gram(–) Bacteria			Fungi			
	A	B	C	D	E	F	G	H	I	J
**1**	128	128	>128	>128	>128	64	>128	>128	>128	>128
**2**	16	32	32	32	>128	>128	>128	>128	>128	>128
**3**	>128	>128	>128	>128	>128	>128	>128	>128	>128	>128
**4**	128	>128	>128	>128	>128	>128	>128	>128	>128	>128
Ampicillin	0.07	0.13	0.13	0.13	>128	16				
Tetracycline					0.5					
Amphotericin B							0.5	1	1	1

A: *Staphylococcus aureus* ATCC25923, B: *Enterococcus faecalis* ATCC19433, C: *Enterococcus faecium* ATCC19434, D: *Salmonella enterica* ATCC14028, E: *Klebsiella pneumoniae* ATCC10031, F: *Escherichia coli* ATCC25922, G: *Candida albicans* SC5314, H: *Aspergillus fumigatus* HIC6094, I: *Trichophyton rubrum* NBRC9185, J: *Trichophyton mentagrophytes* IFM40996.

**Table 2 marinedrugs-17-00133-t002:** List of oligonucleotides used.

Primer Name	Sequence
*CPH1*-For	5′-GAAATGTGGCGCCGATGCAA-3′
*CPH1*-Rev	5′-ACCCGGCATTAGCAGTAGAT-3′
*EFG1*-For	5′-ACAGGCAATGCTAGCCAACA-3′
*EFG1*-Rev	5′-GCAGCAGTAGTAGTAGCAGC-3′
*GAP1*-For	5′-TTAAGTACTGGTGGACCAGC-3′
*GAP1*-Rev	5′-CAAACCCACTTTGAGCAAC-3′
*HWP1*-For	5′-GTGACAATCCTCTCAACCT-3′
*HWP1*-Rev	5′-GAGAGGTTTCACCGGCAGGA-3′
*ALS3*-For	5′-CCACTTCACAATCCCAT-3′
*ALS3*-Rev	5′-CAGCAGTAGTAGTAACAGTAGTAGTTTCATC-3′
*GPD1*-For	5′-AGTATGTGGAGCTTTACTGGGA-3′
*GPD1*-Rev	5′-CAGAAACACCAGCAACATCTTC-3′

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
