# Peer review of "2-Alkyl-4-hydroxyquinolines from a Marine-Derived Streptomyces sp. Inhibit Hyphal Growth Induction in Candida albicans"

_marinedrugs, 2019, doi:10.3390/md17020133_

Reviewer 1 Report

General impression

The authors present a solid study on the isolation and characterization of four antimicrobial compounds from a new species of Streptomyces. One of the compounds was found to have a selective inhibitory action on hypha formation by the dimorphic fungus Candida albicans. The authors follow up with a morphological and molecular characterization of the hypha-inhibiting effect. The experimental procedures are well-described, provide valid results and appear to be statistically sound. However, the authors conclude that the effect of the compound is downstream of Efg1 signaling -  a conclusion that is not fully supported by the data. A more thorough discussion of Efg1/Cph1 signaling is required as it will change the conclusion significantly.

Problems to be fixed

The authors seek to characterize the molecular effects of the 2-alkyl-4-hydroxyquinoline on filamentous growth in Candida albicans through reverse-transcriptase PCR. They describe how probes for key regulators of filamentous growth - EFG1 and CPH1 - do not show changes in gene expression, while probes for downstream targets - HWP1 and ALS3 - show an inhibition of gene expression by the compound. The authors conclude that the effect of the quinolines is downstream of Efg1/Cph1 action. This conclusion is not fully warranted. Efg1 and Cph1 are transcription factors whose activity is regulated on a post-transcriptional level (e.g. by PKA-phosphorylation of a critical residue in Efg1; Bockmuehl and Ernst, 2001. Genetics 157(4), 1523-1520). Previous studies have examined the role of Efg1 and Cph1 in the repression of filamentous growth by farnesol, a related compound. It has been found that even though these proteins are clearly involved in mediating the farnesol effect, there is no significant change in Efg1 or Cph1 mRNA (Langford et al., 2013. Eukaryot. Cell 12(9), 1281-1292; Kebaara et al., 2008. Eukaryot. Cell 7:980–987). All three cited articles conclude that post-translational regulation plays a major role in Efg1/Cph1 signaling. It follows thus that it cannot be ruled out that Efg1 and Cph1 participate in the inhibitory cascade solely on the fact that there is no change in expression of these genes. This point needs to be addressed in the discussion, and the conclusion needs to be changed. 

Minor issues

Please explain why two different strains of C. albicans were used in the study (ATCC10231 for sensitivity, SC5314 for hypha induction). Figures 1 and 4 were cropped in my copy, but I am assuming that they show what the legend states (it looks like the figures were too large for their windows).

Author Response

Response to Reviewer 1 Comments

Point 1: The authors conclude that the effect of the compound is downstream of Efg1 signaling-a conclusion that is not fully supported by the data. A more through discussion of Efg1/Cph1 signaling is required as it will change the conclusion.

Response 1: We highly appreciate the reviewer’s valuable comments. We conclude that the effect of the quinolines is downstream of Efg1/Cph1 action. However, this conclusion is not fully warranted. Accordingly, “downstream” is deleted in the manuscript. We described a more through discussion of Efg1/Cph1 signaling and changed the conclusion (page 7, line 207-217), and added relevant references (page 10, line 408-page 11, line 415) as follows:

“Efg1 and Cph1 are transcription factors whose activity is regulated on a post-transcriptional level. Threonine-206, a phosphorylation site for protein kinase A (PKA) within an Efg1p domain, is essential to promote hyphal induction by environmental factors [25]. Previous studies have examined the role of Efg1 and Cph1 in the repression of hyphal growth by farnesol, a related compound. It has been found that even though these proteins are clearly involved in mediating the farnesol effect, there is no significant change in EFG1 or CPH1 mRNA [26,27]. In our results, EFG1 mRNA levels are regulated during hyphal development, but they were not affected by compound 1, since the magnitude of the changes were similar in the presence and absence of compound 1 (Fig. 4A). These results are consistent with those of Kebaara et al. [26]. Together with our results, this suggests that compound 1 does not regulate EFG1 mRNA levels, but at this time, we cannot exclude the possibility that post-translational regulation of Efg1 is affected by compound 1.”

Point 2: Explain why two different strains of C. albicans were used in this study (ATCC10231 for sensitivity, SC5314 for hypha induction).

Response 2: We routinely used ATCC10231 for sensitivity test and SC5314 for molecular biology experiments. We also already evaluated antifungal activity of the isolated compounds against C. albicans SC5314. These compounds did not show inhibitory activity against both strains (MIC> 128 μg/mL). Following the reviewer’s comment, we revised Table 1 (ATCC1023 is replaced to SC5314).

Point 3: Figures 1 and 4 were cropped in my copy.

Response 3: We also checked the uploaded PDF file and word file in Marine Drugs submission system. As the reviewer’s comments, Figures 1 and 4 were cropped in PDF filed because two figures were too large, but word file is fine. However all Figures and Tables were suitably showed in both our word file and PDF file converted from word. To improve the problem about cropped figures, we adjusted Figure sizes.

Reviewer 2 Report

The manuscript describes the finding of new compounds with hyphal growth inhibition activity. The manuscripr is well written and balanced.

A few major issues should however be addressed:

 - the use of RT-PCR is not quantitative. RT-qPCR should be conducted to assure rigorous results.

 - a simple virulence test, for example using G. mellonella. should be conducted to verify the potential impact of the author's findings. 

Minor issues:

line 64 - why not call the strain S. luridiscaibiei?

figure 3 - it would be nice to see the OD variation at 37ºC to verify if there is any overall growth inhibition.

line 206 - do the authors mean Efg1 is the target of the compounds tested? Otherwise I fail to see where does donwstream means.

Author Response

Response to Reviewer 2 Comments

Point 1: The use of RT-PCR is not quantitative. RT-qPCR should be conducted to assure rigorous results.

Response 1: We sincerely apologized for our incorrect words in the manuscript. In this study, we carried out semi-quantitative RT-PCR to evaluate mRNA expression level of genes related to hypha-inducing signaling pathways. Thus, following the reviewer’s comment, “RT-PCR” is replaced to “semi-quantitative RT-PCR” in the revised version.

Point 2: A simple virulence test, for example using G. mellonella should be conducted to verify the potential impact of the author's findings.  

Response 2: We highly appreciate the reviewer’s valuable comments. The larvae of Galleria mellonella (also known colloquially as the wax worm) is increasingly being used as an infection model to study virulence factors and pathogenesis of many prominent bacterial and fungal human pathogens. When compared with traditional mammalian model hosts, invertebrate infection models are cheaper to establish and maintain. Unfortunately, at present, we have not this infection model system to study C. albicans virulence test. We would like to use this system to our ongoing experiments.

Point 3: Line 64 - why not call the strain S. luridiscaibiei?

Response 3: The reviewer’s concern may need explanation. We analyzed 16S rDNA sequencing of the bacteria strain and the identified sequences compared to S. luridiscabiei. As a result, percentage of identity showed 99% but not 100% in BLAST database. Thus we named the strain as Streptomyces sp. MBTG13 (GenBank accession number: MK408429). If two strains show 100% identity of 16s rDNA sequence, housekeeping genes of S. luridiscabiei would be compared with Streptomyces sp. MBTG13 for detail identification. Further analysis is under consideration.

Point 4: Figure 3 - it would be nice to see the OD variation at 37ºC to verify if there is any overall growth inhibition.

Response 4: Following the reviewer’s concern, we tried to quantification of the effects of compounds 14 on hyphal formation in C. albicans at 37ºC by OD absorption. The deviation of results was very severe due to the hyphal length of C. albicans cells for each sample after 4 h of cultivation was not equal. Therefore, spectrophotometric method was unsuitable except for yeast form. In this study, we determined the morphology of the cells by light microscopy at each time point. A minimum of 200 cells were counted for each sample. Assays were carried out three times, each with three replicates.

Point 5: Line 206 - do the authors mean Efg1 is the target of the compounds tested? Otherwise I fail to see where does donwstream means.

Response 5: We highly appreciate the reviewer’s valuable comments. We conclude that the effect of the quinolines is downstream of Efg1/Cph1 action. However, this conclusion is not fully warranted. Accordingly, “downstream” is deleted in the manuscript. We described a more through discussion of Efg1/Cph1 signaling and changed the conclusion (page 7, line 207-217), and added relevant references (page 10, line 408-page 11, line 415) as follows:

“Efg1 and Cph1 are transcription factors whose activity is regulated on a post-transcriptional level. Threonine-206, a phosphorylation site for protein kinase A (PKA) within an Efg1p domain, is essential to promote hyphal induction by environmental factors [25]. Previous studies have examined the role of Efg1 and Cph1 in the repression of hyphal growth by farnesol, a related compound. It has been found that even though these proteins are clearly involved in mediating the farnesol effect, there is no significant change in EFG1 or CPH1 mRNA [26,27]. In our results, EFG1 mRNA levels are regulated during hyphal development, but they were not affected by compound 1, since the magnitude of the changes were similar in the presence and absence of compound 1 (Fig. 4A). These results are consistent with those of Kebaara et al. [26]. Together with our results, this suggests that compound 1 does not regulate EFG1 mRNA levels, but at this time, we cannot exclude the possibility that post-translational regulation of Efg1 is affected by compound 1.”

Round  2

Reviewer 2 Report

The authors modified the paper according to most suggestions, turning it suitable for publication.